# An Experimental Evaluation of Hemp as an Internal Curing Agent in Concrete Materials

**DOI:** 10.3390/ma16113993

**Published:** 2023-05-26

**Authors:** Rahnum T. Nazmul, Bre-Anne Sainsbury, Safat Al-Deen, Estela O. Garcez, Mahmud Ashraf

**Affiliations:** 1School of Engineering, Deakin University, 75 Pigdons Rd, Waurn Ponds, VIC 3216, Australia; breanne.sainsbury@deakin.edu.au (B.-A.S.); mahmud.ashraf@deakin.edu.au (M.A.); 2School of Engineering and Information Technology, The University of New South Wales, Canberra, ACT 2600, Australia; s.al-deen@unsw.edu.au; 3Arup, Sky Park, 1 Melbourne Quarter 699 Collins St, Docklands, VIC 3008, Australia; estelagarcez@gmail.com

**Keywords:** absorption, carbon negative, desorption, hemp shive, internal curing

## Abstract

The construction industry is facing increased demand for adopting sustainable ‘green’ building materials to minimise the carbon footprint of the infrastructure sector to meet the United Nations 2030 Sustainability Goals. Natural bio-composite materials such as timber and bamboo have been widely used in construction for centuries. Hemp has also been used in different forms in the construction sector for decades for its thermal and acoustic insulation capability owing to its moisture buffering capacity and thermal conductivity. The current research aims to explore the possible application of hydrophilic hemp shives for assisting the internal curing of concrete materials as a biodegradable alternative to currently used chemical products. The properties of hemp have been assessed based on their water absorption and desorption properties associated with their characteristic sizes. It was observed that, in addition to its excellent moisture absorption capacity, hemp released most of its absorbed moisture into the surroundings under a high relative humidity (>93%); the best outcome was observed for smaller hemp particles (<2.36 mm). Furthermore, when compared to typical internal curing agents such as lightweight aggregates, hemp showed a similar behaviour in releasing its absorbed moisture to the surroundings indicating its potential application as a natural internal curing agent for concrete materials. An estimate of the volume of hemp shives required to provide a similar curing response to traditional internal curing techniques has been proposed.

## 1. Introduction

Sustainability is of prime importance in the construction industry since building materials are a significant source of greenhouse gas emissions [1,2]. Concrete, in particular, is responsible for significant greenhouse gas emissions as a result of using cement as a binding agent. As such, the use of ‘green’ building materials can significantly reduce adverse impacts on our environment. Significant research has been reported highlighting numerous examples of using recycled products to replace typical materials in the construction industry [3]. The use of plant-based materials has also been reported to be an effective substitute for chemicals in concrete construction. For example, the application of abaca fibre [4], kenaf cellulose microfiber [4] and eucalyptus fibre [5,6] to improve cracking and shrinkage properties; use of rice husk ash as supplementary cementitious materials to achieve ultra-high performance [7] and use of green artificial aggregate for curing [8].

A recent study [9] provided a detailed review on the multifaced application of hemp toward the sustainable development of the world. The use of hemp in construction is one of them. The use of hemp in concrete (hempcrete) is gaining popularity [10,11]. Hempcrete is a mixture of lime and hemp shives, and is considered to have a low environmental impact [12]. Hempcrete, through the carbon sequestration process, stores carbon in it in a stable form and thus brings the net Green House Gas (GHG) emission to negative values during its manufacture and installation in building products [13]. Hemp tree stem has a highly porous anatomical structure allowing the absorption of large amounts of water with little change to the volume [14]. They are also considered hygroscopic, and, as such, hemp-based concrete products (hempcrete) are often used in buildings to control moisture [15,16,17]. Herein, we propose that the hygroscopic behaviour of hemp could also be used to act as an internal supplier of water within a concrete mass during curing/hydration, and thus could increase the early age strength of concrete.

In the process of internal curing, water stored in curing agents are utilized by the cementitious materials to expedite chemical reactions [18]. The technique of internal curing is beneficial to scenarios where conventional (external) curing methods are difficult due to accessibility or site constraints [19] or in dense concrete with low w/c ratio [20]. Internal curing can also be used to prevent issues with external curing due to evaporation of water from the cement paste, which, in turn, may promote shrinkage cracking of concrete [21]. Consequently, the overall strength and durability of the concrete structure could be adversely impacted [19]. Insufficient curing limits the hydration process resulting in lower early strength development. This issue is more critical in temporary support products such as shotcrete [22] where rapid strength gain is required, which is instigated by using chemical accelerators. Due to the rapid hardening, the development of early age cracks makes shotcrete more vulnerable to physical and chemical attacks affecting its durability. Providing an additional internal curing agent could reduce early age cracking and significantly improve the early age strength gain and durability [23].

Apart from the conventional internal curing agents that are manufactured such as lightweight aggregates (LWA) and Superabsorbent Polymers (SAP), the performance of plant-based agents has also been considered [24] in the forms of cellulose, eucalyptus, kenaf, and hardwood pulp fibers [4,25,26]. Previous studies on plant based internal curing materials showed some encouraging results. However, to achieve a sustainable solution, the potential of plant-based internal curing materials must not be limited by local geographical regions. Thus, additional research is required to find other plant-based alternatives that are available in abundance worldwide and are a hardy, tolerant and annual plant species such as hemp which is ubiquitously grown and distributed globally as a food, medicine, or fabric [27]. Hemp shives—the chipped pieces of porous hemp tree stem possess excellent water absorbing characteristics, about 2–3 times of its own weight [28]. This hygroscopic behaviour shows potential for its use as an ideal candidate for a sustainable plant-based internal curing agent. Current research examines the possibility of utilizing hemp shives as an internal curing agent through material and mechanical characterization of Australian grown hemp.

### 1.1. Properties of an Internal Curing Agent

In general, when a concrete batch is prepared and placed, the relative humidity (RH) starts to drop, as the mixing water is used, and self-desiccation commences. When an internal curing agent is applied, stored water within the agent gets transferred to the cement paste with a reduction in the RH. With a small RH drop, water in larger pores will have less capillary tension and should serve the cement paste faster. Gradually, as hydration progresses, with the reduction in pore size, water stored in smaller sized pores will flow into the cement paste with a further decrease in RH.

Thus, to achieve the main objectives of internal curing, an agent should release its absorbed moisture at a high relative humidity at the required time. Previous research has shown that the ideal internal curing agent should release at least 85% of its absorbed water at 94% relative humidity [29,30]. On the other hand, a less efficient internal curing agent holds its absorbed water and releases it only gradually with the decrease in relative humidity. The characteristic factors to determine the efficiency of an internal curing agent were reported in [25] and include:

Water saturation status: it is essential to ensure that the internal curing agent is fully saturated by either soaking in water for 24 h prior to mixing (i.e., porous aggregate and luminous cellulose fibres) or by adding water during the mixing phase (i.e., superabsorbent polymer).

Water absorption and desorption properties: This agent response depends on the pore structure of the material and interaction with the cement paste. Water is required to be released at the onset of early self-desiccation of the cement paste. An earlier release may affect the rheological characteristics of the cement paste rather than aiding in internal curing [31]. Furthermore, water migration is only possible from the pores of the internal curing agent to the cement paste when its pore size is larger than the equivalent pore radius of the cement paste. Thus larger and open pores of the internal curing agent are better as it can hydrate cement paste with finer pores [32].

Particle size distribution: Cement paste located within the migration distance of the internal curing agent water when released is mitigated from self-desiccation. As such, the particle size distribution of the internal curing agent should be well distributed inside the cement paste to ensure greater surface area contact [33]. In the case of pumice aggregate, a particle size ranging between 1.18–2.36 mm was reported to perform better than 0.6–1.18 mm in reducing autogenous shrinkage [34]. A similar finding was observed in case of rice husk ash where a particle size of 5.6 µm and 9.0 µm performed better than 3.6 µm [35]. However, a different finding has been noted in case of Kenaf fibre [4] where small sized particles (400 µm) performed better in bridging micro-cracks and thus exhibited a better compressive strength compared to larger particle size (5 mm) when added at the same dosage.

### 1.2. Hemp Shives: An Internal Curing Agent

Although the infusion of plant fibres in concrete materials for increasing the tensile strength and toughness of concrete dates back to the end of the previous century [36,37], the idea of using plant fibers for internal curing is contemporary [25]. Plant fibers that have been previously considered as an internal curing agent include modified kenaf [4,26,38], hardwood pulp [39], luffa [6], eucalyptus [5], thermomechanical pulp [40], and cellulose [41]. Research has revealed that the natural plant fibres are able to contribute to the internal curing of concrete in a similar manner to SAP [42] and LWA [43]. The long-term durability of these lignocellulosic agents is a matter of concern, however, relevant research has shown that plant-based agents positively contribute to minimise self-desiccation and autogenous shrinkage during the early days of hydration before they degrade [7]. Herein, we propose using hemp shive for the internal curing of cementitious material based on its morphological characteristics such as pore size distribution, physical properties as absorption and desorption behaviour with varying particle sizes.

### 1.3. Pore Size Distribution

Hemp shives are porous in structure, consisting of interconnected voids with a very low density. As such, they are highly absorbent and capable of retaining significant amounts of water [10]. Pore sizes can be divided into gel nano pores (<0.01 µm), mesopores (0.01–0.05 µm), middle capillary pores (0.05–0.1 µm), large capillary pores (0.1–5 µm), and macro pores (>5 µm). [44,45].

A detailed study suggests an average pore radius varies from 0.03 to 80 µm [14] and the total accessible porosity in hemp shive is in the order of 76.67 ± 2.03%. Presence of micro-pores (0.03 to 1 µm) in the pit membrane, cell walls and pit aperture were confirmed through SEM image analysis, whilst macro pores (20 to 80 µm) were present in the vessels. Voids in the parenchyma cell vary from 1 to 20 µm. Because of this unique microstructure, a multi-scale porosity is observed which is not subjected to change even after cutting and retting. It should be noted that retting is the process used to separate the hemp fibres from the stem, and is conducted with the help of microorganisms, acid/bases, and/or enzymes. A recent study [46] suggests that retting increases the water absorption property of hemp shives by 20%. As part of the current study, pore size distribution of hemp shives was compared against those reported for Expanded Lightweight Aggregate (ELWA), Recycled Aggregate, and Expanded shale as shown in Figure 1. Here, the *x*-axis represents the size of the pores, and the *y*-axis represents the volume of mercury required for filling the pore under different pressure for a specimen in Mercury Intrusion Porosimetry (MIP) method. It is observed that, with respect to pore size distribution, hemp shives have a greater number of pores with larger diameters when compared to other internal curing agents. It is worth noting that the proportion of inaccessible pores (<3 nm) present in hemp shives are lower than that of hemp fibre. Hemp shives possess a wide variation in pore size distribution compared to fibres [47] and, thus, should absorb significantly higher amount of water, which may be beneficial to curing.

Previous research showed that a curing agent loses its effectiveness in internal curing if the pore sizes are smaller than 100 nm, which makes it difficult in releasing the stored water [50]. Of the traditional internal curing agents, ELWA is the most efficient since 89.7% of its pores are greater than 100 nm. With slight variation, the micropores present in the pit membrane, cell wall, and pit aperture of hemp shives are predominantly above 100 nm in size [47]. Moreover, they have the added benefit of significantly larger pores than ELWA, and hence, there is a strong likelihood for application as an internal curing agent in concrete.

### 1.4. Absorption Behaviour

Being highly porous, hemp shive exhibits excellent water absorption behaviour [28,46,47,51,52]; test results showed that shives are capable of absorbing water equivalent to two-thirds of their own weight within 10 min of immersion reaching a saturation level of 95% [28]. This behaviour can be attributed to the presence of the macro pores due to the hemicellulose and lignin content. Presence of larger macro pores in hemp shive allows it to absorb water at a faster rate than a smaller pore size that slowly fills up with water [28,46,47,51,52]. Properties of other common internal curing agents such as LWA, which have smaller pore structure, suggest they will continue to absorb water for three days, but will likely reach their maximum after 48 h [29].

### 1.5. Desorption Behaviour

Although the water absorption property of hemp has been investigated by several researchers, very limited research has been reported on its water releasing behaviour, which is important for internal curing. A recent study analysed the moisture and heat sorption properties of hemp fibre and shive [47]. Both the hemp fibre and hemp shive were subjected to successive sorption and desorption over three consecutive cycles at a constant temperature and were plotted against relative humidity (RH). It was observed that in both cases for a single cycle, both the sorption and desorption curves from shives formed a hysteresis loop where the rate of water absorption (0.57–0.66) was higher than the desorption rate (0.27–0.35) at a RH range of 80–90%. This means, at high RH range hemp shives absorb water at a quicker rate compared to the rate of release during desorption.

The hemp shives’ hysteresis behaviour can be explained by the presence of micropores and the permanent uptake of water through pore molecules as interpreted by Collet, Bart [16]. In addition, capillary condensation hysteresis, contact angle hysteresis, and an ink-bottle effect may also contribute to such properties [16]. Research on eucalyptus pulp showed that within 25 h of hydration, water stored in the pulp lumen was released to the self-desiccating cement paste as a result of relative humidity gradient [25]. In the same way, water should be released from larger pores of hemp first and then gradually from smaller pores to promote internal curing of the cement paste. As hydration progresses, larger pores are divided into progressively smaller pores. This gradual water migration from the meso-pores to the smaller capillary pores should serve the hydrating cement paste. Research [32] shows that capillary pores in the range of 5–50 nm are more susceptible to cause capillary stress resulting in autogenous strain. Pore sizes in the pit membrane, aperture and cell walls of hemp shives are in the range of 0.03 µm to 1 µm and thus should help in mitigating autogenous shrinkage.

### 1.6. Particle Size Distribution

The particle size of the curing agent is particularly important to ensure better dispersion of the curing water into the matrix. This has previously been understood through previous research, such as the improvement of compressive strength and autogenous shrinkage properties of concrete though the addition of kenaf fibres [4]. This benefit is provided by the introduction of additional internal curing water through increased fibre dosage. In this regard, shorter fibres have been found to perform better in shrinkage mitigation at early ages which is largely due to their better dispersion into the matrix. To find out the effect of size variation of hemp shives in internal curing of concrete, a balance between the particle size distribution, water absorption and desorption capacity, and necessary dosage determination is required.

## 2. Materials and Methods

### 2.1. Materials

Hemp shives, produced and processed by Australian Hemp Manufacturing Company a subsidiary of Developing Sustainable Direction Pty Ltd. (DSD) in Victoria, were considered in the current study in an untreated condition. The shives were washed under potable flowing water to remove dust particles on the shive surface and oven dried for 48 h prior to conducting the laboratory procedure. Figure 2 shows the commercially available hemp shives studied herein.

### 2.2. Experimental Method

#### 2.2.1. Particle Size Distribution and Morphological Analysis

A particle size distribution analysis of the hemp shives samples was conducted using both traditional sieve analysis (according to ASTM C136/C136M [53]) and image analysis. The resulting particle size gradation curve (Figure 3) shows that the shive sizes varied widely between 30 mm and 0.5 mm although most of the particles (>70%) were between 1 and 10 mm. Based on the results of sieve analysis, three size ranges of hemp shives, as shown in Figure 3, have been selected for further analysis such as HS1 (fine) 1.18 to 2.36 mm, HS2 (medium)—2.36 to 4.75 mm and HS3 (coarse)—4.75 to 6.70 mm.

A two-dimensional image analysis was performed on representative samples from each of the selected size ranges to obtain a better understanding on the aggregate size distribution. This method is mainly applied for non-spherical particles for size distribution and morphology determination [54], particularly in plant based aggregates. Different from sieve analysis, where particles are separated based on their width, image analysis provides specific information associated with particle shape and morphology. The software ImageJ freeware (version 1.51j8) [55] was used for the image analysis according to RILEM TC 236-BBM [56] protocols. A sample size was selected that corresponds to a minimum of 1000 particles [56]. The samples were spreaded over a black background for optimum image contrast. The image was acquired with a digital camera with 72 dpi resolution and a binary image was produced from the colour image. Each object outline was picturized on the image after spatial calibration and is presented in Figure 4.

The particle shape analysis of each of the hemp shive grades (fine, medium, coarse) is shown in Figure 5. The analysis suggests that the length-to-width ratio, i.e., aspect ratio of the hemp shives is consistent regardless of their size range.

#### 2.2.2. Density and Initial Moisture Content of Hemp Shives

Bulk density and initial moisture content in each of the defined hemp shives samples (fine, medium, and coarse) was determined following the procedure described by Amziane, Collet [56]. It is important to determine the bulk density of hemp shive aggregates as it indicates the volume of the concrete replaced by the shives in the mix. The weight of the aggregate is also dependent on the initial moisture content of the shive, and hence, the initial moisture content of HS1, HS2, and HS3 categories were measured. The bulk density ρ (kg/m^3^) was measured by weighing the aggregate and measuring the corresponding water volume.

Firstly, the sample was dried in an oven at 60 °C till a constant mass was reached. Then, the dry sample was used to fill up half the volume of a glass cylinder with known mass. Next, the sample was mixed well by upending the cylinder 10 times. Then, the cylinder was shaken to create a horizontal surface and the surface height was marked. The mass of the cylinder plus the sample was taken to determine the hemp shive mass for bulk density. The volume of the shive was then replaced from cylinder by same volume of water which provided the bulk volume of the sample. Then, the bulk density was calculated from Equation (1) and three measurements were made to provide an average result.
(1)$$\rho =\frac{M}{V}$$
where,
*ρ* = bulk density, kg/m^3^*M* = mass, kg*V* = bulk volume, m^3^


Representative 200 g samples were used to determine the moisture content according to [56]. The initial moisture content was measured by weighting the aggregates in its initial state and after drying at a temperature at 60 °C for 48 h. The initial moisture content of the hemp shives was calculated based on Equation (2) and obtained results are summarised in Table 1.
(2)$$W=\frac{M_{i}-M_{d}}{M_{d}}\times 100,$$
where,
*W* = initial moisture content, %*M_i_* = initial mass of the sample, kg*M_d_* = oven-dried mass of the sample, kg

**Table 1 materials-16-03993-t001:** Properties of hemp shive aggregates.

Properties	HS1	HS2	HS3
Initial moisture content (% weight of oven dried sample)	8.7	8.2	9.5
Bulk density (kg/m^3^)	86.76	113.85	121.92
Water absorption after 72 h (% weight of oven dried sample)	335.6	321	296

The Coefficient of Variation was less than 5%.

#### 2.2.3. Water Absorption Behaviour

Water absorption properties of Australian hemp shives used in the current study were determined following the guidelines proposed in Amziane, Collet [56]. The water absorption properties of each hemp shive sample HS1, HS2, and HS3 was measured for 72 h following the methods outlined in [29,48,57] for angular particles such as hemp shives (Figure 6). The water absorption of shive samples at a specific time was calculated based on Equation (3) and obtained results are summarised in Table 1.
(3)$$\mathrm{Absorption}\;(\%)=\frac{m_{wet}-m_{dry}}{m_{dry}}\times 100,$$
where,
Absorption = water absorption of hemp, %*m_wet_* = mass of wet sample, kg*m_dry_* = mass of oven-dried sample, kg


**Figure 6 materials-16-03993-f006:**
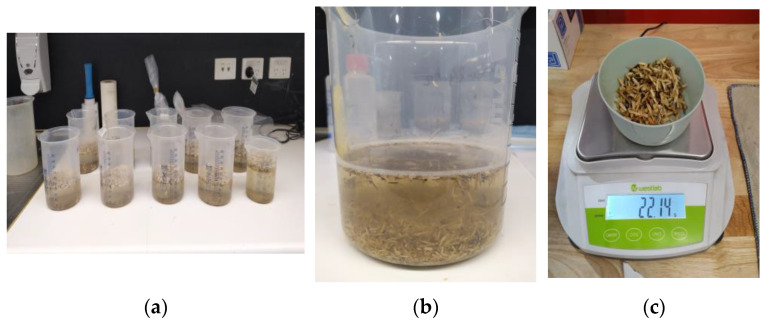
Experimental images of determination of hemp shive properties (**a**) samples for water absorption and desorption test (**b**) saturated sample at 72 h (**c**) weight measurement of saturated sample in saturated surface dry condition.

#### 2.2.4. Moisture Desorption Behaviour

The process of water release from hemp shives during drying is measured through a desorption isotherm, which represents the mass loss as a function of relative humidity at a constant temperature. The desorption behaviour of hemp shives was monitored by measuring the amount of water loss from saturated hemp shives (i.e., RH 100%) when subjected to 93% relative humidity at a constant temperature [58,59]. The change of mass of the sample was measured initially at 5 min, 10 min, 15 min, and 30 min to monitor the desorption pattern at early stages, and then at 1 h, 2 h, 3 h, 4 h, 24 h, and 48 h until the change in the moisture content was less than 0.01%.

The desorption rate was evaluated from 100 to 93% relative humidity as this range is important for internal curing [20]. Hemp samples in its Saturated Surface Dry (SSD) condition were considered to represent sample at approximately 100% RH.

## 3. Results and Discussion

### 3.1. Effect of Hemp Particle Size on Hemp Properties and Water Absorption

It was observed that properties such as initial water content, bulk density, and water absorption capacity of the aggregates had some variation across three categories. The initial moisture content for HS3 samples were considerably higher than HS2 and HS1 samples. The bulk density of the aggregates increased as particle size increased. The bulk density of HS3 was more than 40% of that of HS1. This indicates, when added to concrete for internal curing, coarser shives (HS3) will replace more concrete than finer shives (HS1) when replaced by the same volume, which could eventually affect the mechanical performance of concrete. The results on bulk density are well aligned with the findings from previous research conducted on Australian grown hemp shives [60]. A lower bulk density in the smaller shive particles were supported by the presence of more material fineness. Similarly, the initial water content of the particles showed an increase with size.

Unlike initial moisture content and bulk density, the water absorption capacity of the hemp shive samples in saturation decreases with an increase in particle size. Although coarser shives such as HS3 hold more moisture than finer shives, the latter absorbed more water per unit weight. HS1 has more particle surface area compared to that of HS2 and HS3. A larger surface area may contribute to absorbing more water when compared to larger-sized shives. This property can be found beneficial in terms of internal curing as smaller particles replace less amount of concrete but adds more water in comparison to that of coarser shives such as HS3.

The water absorption properties of each of the hemp shive grades HS1, HS2, and HS3 have been compared with that of traditional internal curing agents in Figure 7. The water absorption of each type of shive has been normalised by its maximum value of saturation. As it is observed from Figure 7, initially all agents demonstrate a high rate of water absorption that gradually decreases prior to reaching the peak saturation point.

Hemp and expanded shale agents, upon reaching a water absorption level of ~65–70%, showed a noticeable reduction in their rate of absorption. The shale reaches 100% capacity after ~24 h, however, hemp shives continue to absorb water for an additional 24 h, albeit at a slower rate. The recycled aggregate reaches 80% saturation within 1 h, 95% within 2 h, and becomes fully saturated within 24 h. From the comparison of hemp shives with expanded shale and recycled aggregate, it is observed that hemp shives absorb water in a similar manner to that of the shale and recycled aggregate which is a positive attribute to be considered for internal curing.

The water absorption rate from an oven dry state to over 48 h was determined for the considered three different hemp shive samples HS1, HS2, and HS3 and obtained results are presented in Figure 8.

Figure 8 shows that both the water absorption capacity and the rate of water absorption decrease with the increase in particle size. The maximum water absorption capacities were observed as 335%, 321%, and 296% for HS1 (fine), HS2 (medium), and HS3 (coarse) samples, respectively. The higher water absorption capacity of the smaller particles may be indicative of the presence of more permeable pore count on the surface of the smaller particles. It is worth noting that HS1 offers the largest surface area among all three shive samples, and this may have contributed to the increased water absorption. A similar finding was observed in the case of recycled aggregates [48]. However, variation in water absorption rate for the considered hemp samples was observed only during the early conditions (0–4 h) but the absorption rates were similar for all samples during the later stages. Figure 9 presents results normalised over 24 h for the considered hemp shive samples.

It is obvious that the hemp shives, irrespective of their sizes, showed a similar water absorption behaviour with time that can be represented using normalised absorption *S* by Equation (4):*S* = *a* ln(*t*) + *b*(4)
where, *S* = the hourly moisture absorption normalized by 24 h moisture absorption, unitless; *a* and *b* = constants depending on hemp absorption behaviour, unitless; *t* = time, hours.

Table 2 shows the values of a and b determined from the corresponding test results for HS1, HS1, and HS3. Relevant R^2^ values are also presented in Table 2 showing good agreement with the proposed fit, which are shown in Figure 10, Figure 11 and Figure 12.

Whilst the predictive models for all samples showed good agreement, HS3 samples produced the best fit which could be attributed to the smaller accumulated surface area of the larger particles resulting in more consistent results. From an absorption perspective, the proposed equation could serve as a useful tool to determine the normalised absorption *S* value for a specific size of hemp.

### 3.2. Moisture Desorption Behaviour of Hemp Shives

The desorption results for each of the three different size ranges (i.e., HS1, HS2, and HS3) of hemp shives are presented in Table 3.

A comparison between water desorption behaviour of hemp shives, expanded shale and LWAs and pine fibre is shown in Figure 13. A common trend is observed in desorption behaviour highlighting an initial steep desorption, followed by a more gradual decrease, and finally reaching a plateau where no more water release occurs.

Hemp shive samples released water in the fastest rate when compared against other curing agents. Among the three different hemp shive sizes, it was observed that HS1 (fine) and HS2 (medium) released ~94% of the stored water within 24 h. The desorption rate for HS3 (coarse) was somewhat slower as it released 88% of its water within 48 h. Expanded slate IC followed a similar trend to hemp shives, releasing approximately 79% of stored moisture within 23 h and almost 94% after 24 h.

To perform efficiently as an internal curing agent, a material should release most of its absorbed moisture at a high relative humidity (i.e., 93%) to facilitate hydration. The moisture desorption behaviours of the hemp shive samples from 100% RH to 93% RH are presented in Figure 14.

As observed from the graph, hemp shives released almost 95% of their absorbed moisture within the first 24 h subjected to a high relative humidity. This rate maximized maximum during the first few hours (<3 h) and decreased gradually over the next 24 h but released most of its absorbed moisture by then. HS1 and HS2 samples with smaller particle size showed significantly higher desorption rate under high RH when compared against that for HS3 samples. Figure 15 shows the normalized (over 24 h) water desorption behaviour for all samples.

The normalised desorption response of hemp shives can be expressed by Equation (5):*D* = *me^n^*^t^,(5)
where,
*D* = the hourly moisture desorption normalized by 24 h moisture desorption, unitless *m* and *n* = constants depending on hemp absorption behaviour, unitless*t* = time, hours.


Table 4 shows the *m* and *n* values for hemp shive samples and the relevant R^2^ values showing good agreement between the proposed curve and the test results. Figure 16, Figure 17 and Figure 18 show the fitting curves of HS1, HS2, and HS3 moisture desorption.

The coarse particles (HS3), such as absorption, produced more consistent results for desorption. It should be noted that desorption behaviour is highly dependent on particle size for the hemp shives.

### 3.3. Analytical Model for Estimation of Hemp Shives for Internal Curing

The current study shows that particle size (aspect ratio), density, and water absorption behaviour of hemp shive samples are almost identical, which paves the way for proposing an analytical model based on their desorption behaviour. The proposed models will be used to determine the appropriate quantity of hemp shive to ensure its efficient use as an internal curing agent in a concrete mix.

An analytical model for LWA is currently available that predicts the mass requirement for the internal curing agent [20,33]. This model was developed considering three important parameters such as required amount of internal curing water movement capability of water and spatial distribution of the internal reservoirs within concrete/mortar.

By equating the demand of water required for hydration with the corresponding supply, the mass requirement of LWAs (*M_LWA_*) may be determined following Equation (6) [33]:(6)$$M_{LWA}=\frac{C_{f}\;\times \;CS\;\times \;\alpha_{max}}{SL\;\times \;\varphi_{LWA}}$$
where,
*M_LWA_* = Mass of required internal curing agent, g*C_f_* = the cement factor of concrete mixture expressed, g*CS* = the chemical shrinkage of the binder at 100% reaction, which for Portland cement is 0.07 mL/g cement [18]*α_max_* = the expected maximum degree of reaction for the binder (0–1), unitless*SL* = the saturation level of curing agent, unitless*φ_LWA_* = the measured sorption capacity of the curing agent, unitless.


It is worth noting that the underlying assumptions of this equation are all water is to be readily available for use of the cement paste [33]; and all voids are considered to be filled with water when saturated.

Contrary to these assumptions, hemp shives possess a different desorption behaviour, and all absorbed water may not be always readily available. As such, the amount of water released by hemp shives could be obtained using Equations (4) and (5), and the proposed Equation (7) may be used to determine the required mass of hemp shives *M_HS_*.
(7)$$M_{HS}=\frac{C_{f}\times CS\times \alpha_{max}}{S\times \varphi_{HS24h}\times D},$$
where,
*M_HS_* = Mass of required hemp for internal curing, g*C_f_* = the cement factor of concrete mixture expressed, g*CS* = the chemical shrinkage of the binder at 100% reaction, which for Portland cement is 0.07 mL/g cement [18]*α_max_* = the expected maximum degree of reaction for the binder (0–1), unitless*φ_HS24h_* = the mass of water per unit weight of hemp shive at 24 h, g*S* = the hourly moisture absorption normalized by 24 h moisture absorption, unitless*D* = the hourly moisture desorption normalized by 24 h moisture desorption, unitless


It should be noted that the parameters of Equation (7) as, *φ_HS24h_* could be determined from Figure 9, *S*, and *D* could be determined using Equations (4) and (5).

Based on the equation proposed, Table 5 estimates the mass of hemp shives required for internal curing per unit volume of concrete based on their particle size, normalised water absorption *S* over 24 h and, normalised moisture desorption *D* over 24 h. It shows that with the increase in particle size, the quantity required for hemp shives increase. For example, HS1 0.8% of the binder mass HS1 should sufficiently meet the internal curing water demand per unit weight volume of concrete while those for HS2 and HS3 are 1.1% and 3.3%, respectively.

The addition of HS1, HS2, and HS3 in a concrete mix will involve a replacement of same volume of concrete. Thus, there will be a reduction in compressive strength due to the reduction in concrete materials within a specified volume. Considering the estimation provided in Table 1, HS1 with its minimum bulk density and minimum percentage required should have a superior performance over HS2 and HS3 in internal curing. Further investigations on this topic will be required for the efficient use of hemp shives as an internal curing agent.

## 4. Conclusions and Future Recommendations

Internal curing is an important process in concrete construction because it improves the mechanical properties (e.g., compressive strength) of the product and makes it more durable by minimising shrinkage cracking which could be critical in the absence of adequate curing. Many internal curing agents are available in the market but the most common is lightweight aggregate (LWAs). LWAs are artificially produced, and the production process releases particulate matter (PM), greenhouses gas, as sulphur, and nitrogen-based oxides. As such, the use of various plant-based fibres (e.g., eucalyptus pulp, luffa and kenaf) are being considered as alternatives. These plant-based fibres have been found to assist in internal curing and reduce autogenous shrinkage. However, they are not all readily available ubiquitously around the globe. As such, the hardy, tolerant, annual plant species hemp is considered.

Hemp has a porous anatomical structure and hygroscopic nature; thus, it can absorb and desorb large amounts of water, which aids internal curing. As such the characterization of Australian hemp shives was undertaken to evaluate its’ performance as an internal curing agent. The shives performance was analysed using three distinct size ranges to produce different products HS1—fine (1.18–2.36 mm), HS2—medium (2.36–4.75 mm), and HS3—coarse (4.75–6.70 mm). Following points could be made based on the experimental works and analyses conducted,

The bulk density of the shives increased with size. It was found to be highest for the HS3 samples and lowest for the HS1 samples.It was found that hemp shive particles can absorb water as high as 335% of their initial weight, however, their capacity decreases with an increase in particle size.The water absorption rate is observed to be fastest at early ages (3 h) reaching 220% of its initial weight before gradually decreasing and reaching a maximum after 48 h.Likewise, the moisture desorption rate was found to decrease with the increasing size of the hemp shive particle. Thus, the smallest particle sizes dissipate most of their absorbed moisture at a relative high humidity of 93%, with the largest particle size releasing moisture more slowly under high humidity conditions, making them less suited for internal curing.During moisture absorption hemp shives were slower to reach the maximum capacity compared to the other internal curing agents.Hemp shives demonstrated excellent qualities during moisture desorption, releasing about 94% of moisture within 24 h (for HS1 and HS2) which is the fastest among the internal curing agents typically used in the industry.The efficiency of hemp shives to supply the required curing water, has been considered through the development of a hemp-specific analytical model based on their size-specific property variation. The proposed model provides an estimation of hemp shive required for internal curing for a cementitious material.

However, further validation of this model by concrete mixes is recommended. As the proposed model is based on the observed moisture desorption behaviour of the particles, a detailed empirical study should be performed to confirm the analytical observation from this study before accepting hemp as an internal curing agent. Hereby, there is significant scope to proceed this study further on the mechanical property development of the hemp cured concrete with the progression of curing age.

## Figures and Tables

**Figure 1 materials-16-03993-f001:**
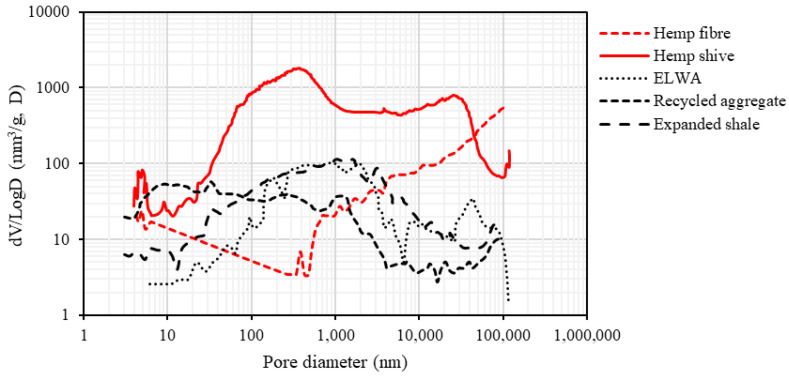
Pore size distribution of hemp shives, hemp fibres, ELWA [23], expanded shale [48], and recycled aggregates [49].

**Figure 2 materials-16-03993-f002:**
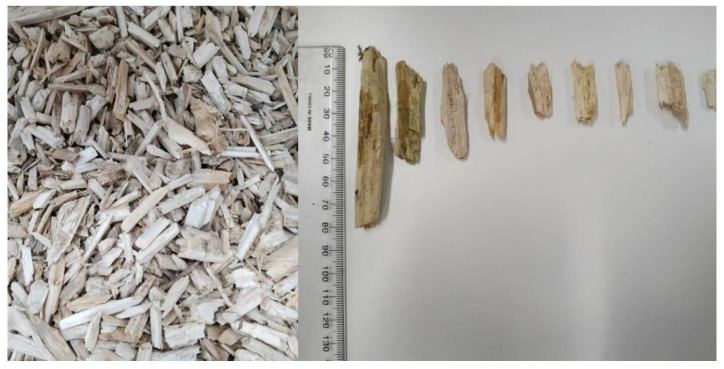
A representative sample of hemp shive purchased from the Australian hemp Manufacturing Company.

**Figure 3 materials-16-03993-f003:**
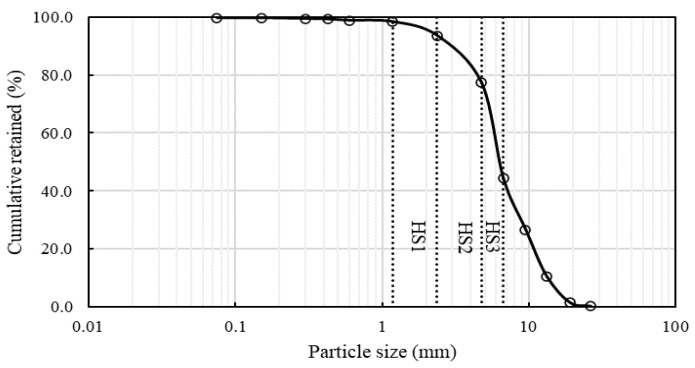
Particle size distribution of a representative sample of hemp shive.

**Figure 4 materials-16-03993-f004:**
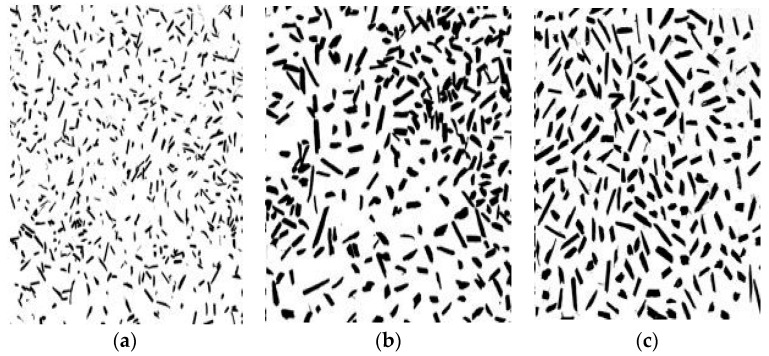
Binary image of aggregate size distribution of HS1 (fine, (**a**)), HS2 (medium, (**b**)) and HS3 (coarse, (**c**)) obtained through a 2D-image analysis technique.

**Figure 5 materials-16-03993-f005:**
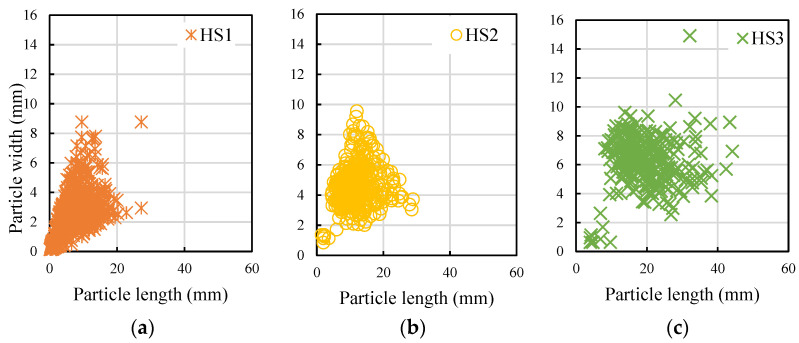
Particle size analysis of HS1 (**a**), HS2 (**b**), and HS3 (**c**) through image analysis technique.

**Figure 7 materials-16-03993-f007:**
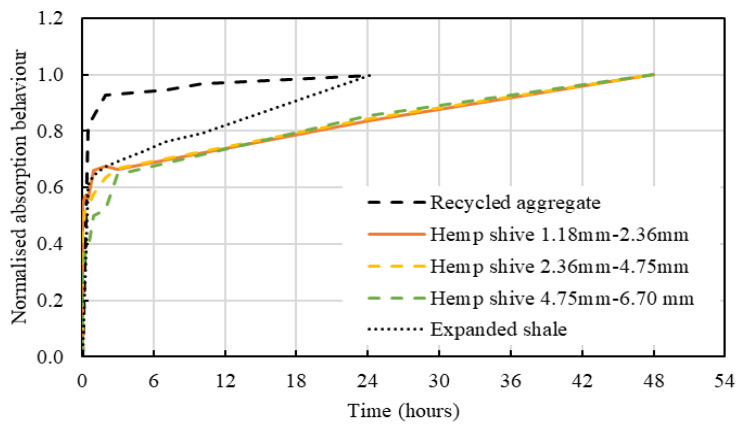
Comparison of water absorption behaviour of hemp shives, expanded shale, and recycled aggregate.

**Figure 8 materials-16-03993-f008:**
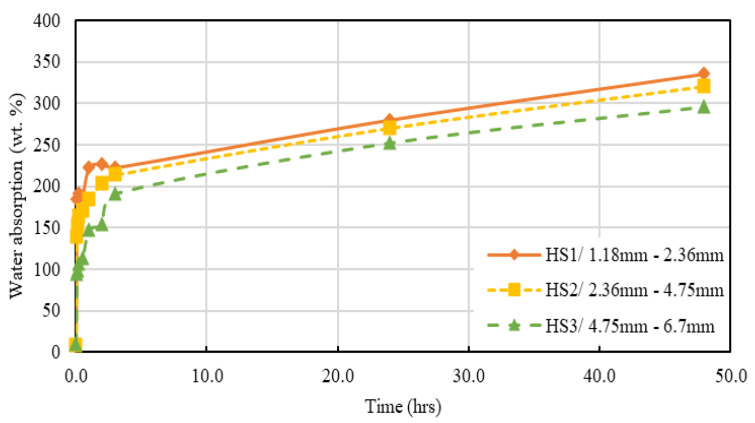
Variation in water absorption behaviour of hemp shive samples HS1, HS2, and HS3.

**Figure 9 materials-16-03993-f009:**
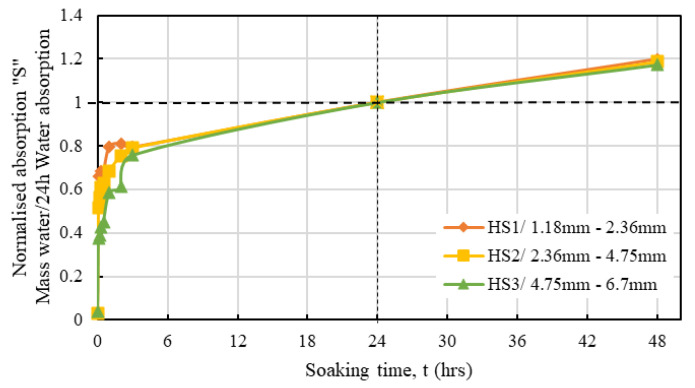
Time-dependent water absorption normalised by 24 h water absorption of HS1, HS2, and HS3.

**Figure 10 materials-16-03993-f010:**
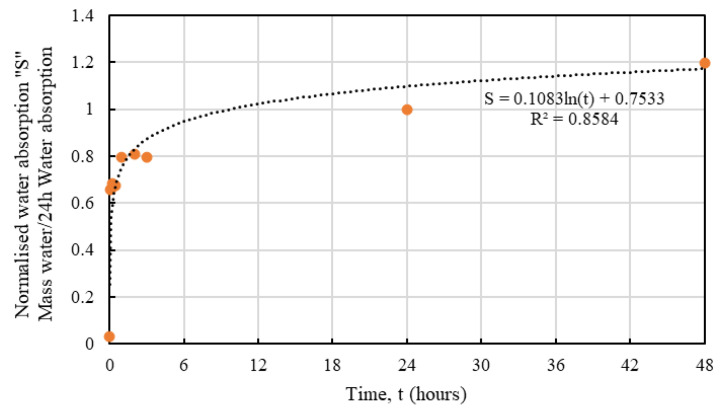
Predictive model for water absorption behaviour of HS1.

**Figure 11 materials-16-03993-f011:**
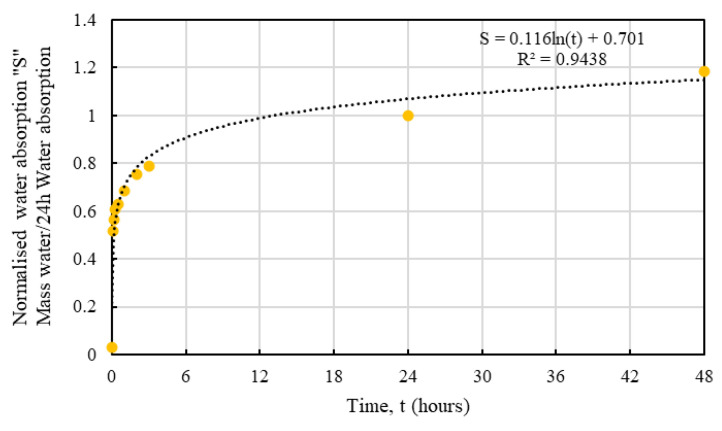
Predictive model for water absorption behaviour of HS2.

**Figure 12 materials-16-03993-f012:**
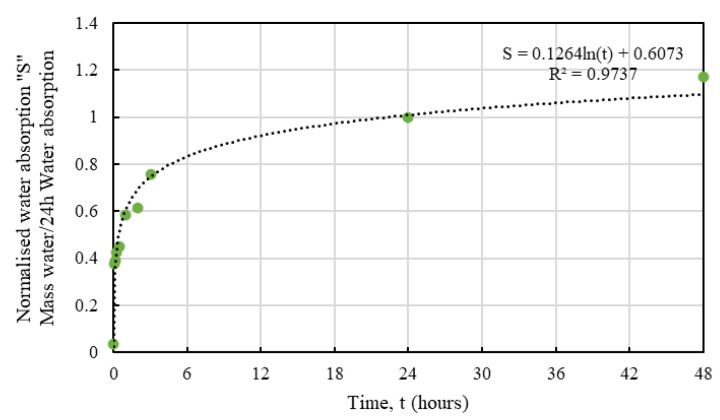
Predictive model for water absorption behaviour of HS3.

**Figure 13 materials-16-03993-f013:**
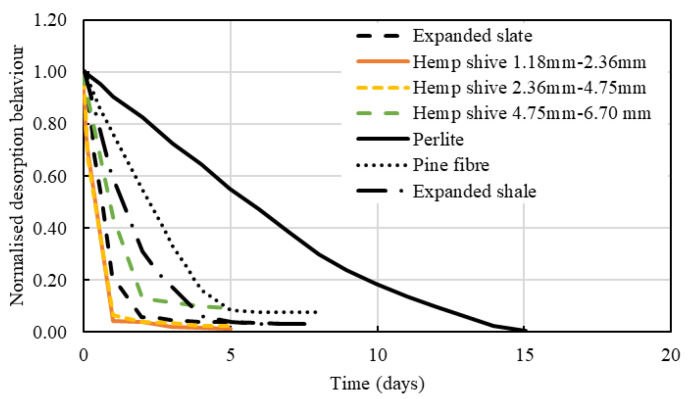
Comparison of water desorption behaviour of hemp shives, expanded shale [48], LWAs [61], and pine fibre [61].

**Figure 14 materials-16-03993-f014:**
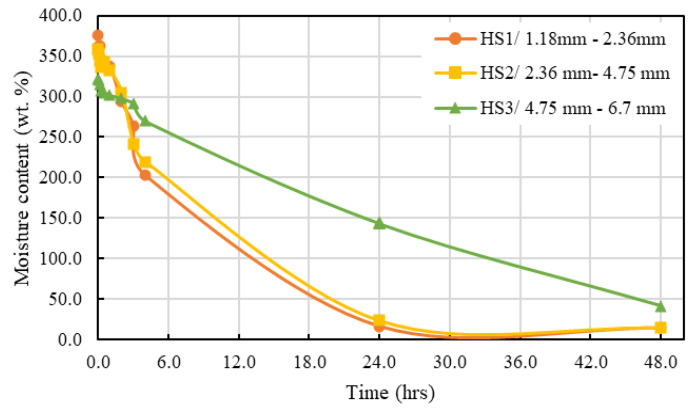
Moisture desorption behaviour of hemp shives from SSD condition (100% RH) to 93% RH.

**Figure 15 materials-16-03993-f015:**
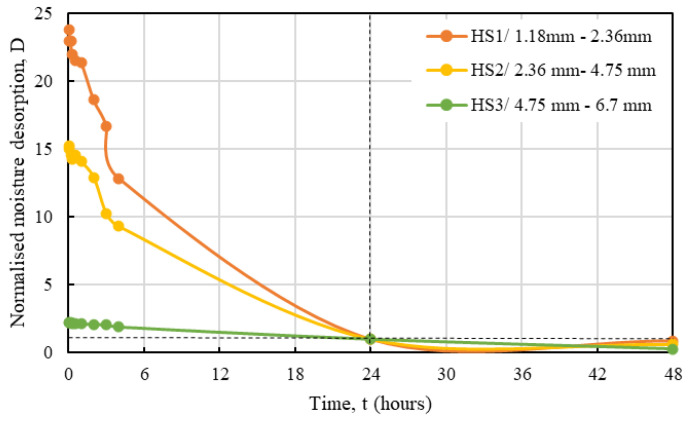
Moisture desorption behaviour of HS1, HS2, and HS3 normalised by 24 h moisture desorption.

**Figure 16 materials-16-03993-f016:**
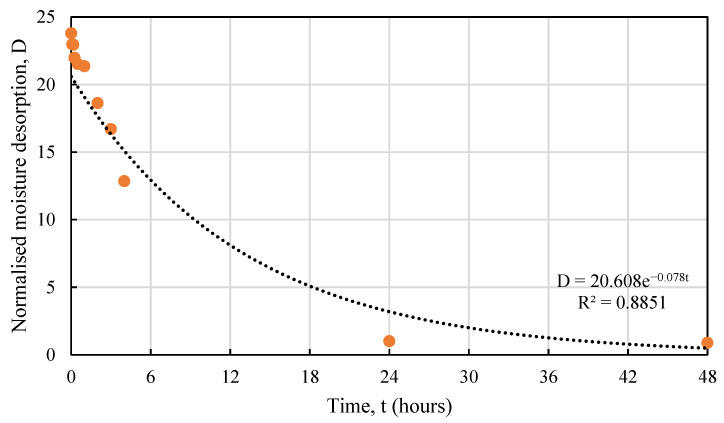
Test vs. proposed relationship for HS1 moisture desorption behaviour.

**Figure 17 materials-16-03993-f017:**
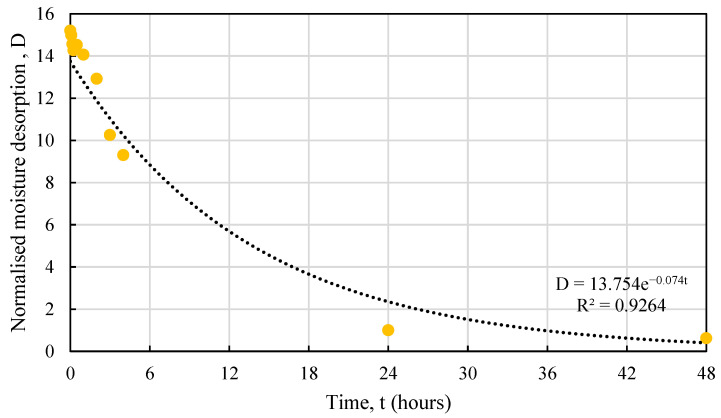
Test vs. proposed relationship for HS2 moisture desorption behaviour.

**Figure 18 materials-16-03993-f018:**
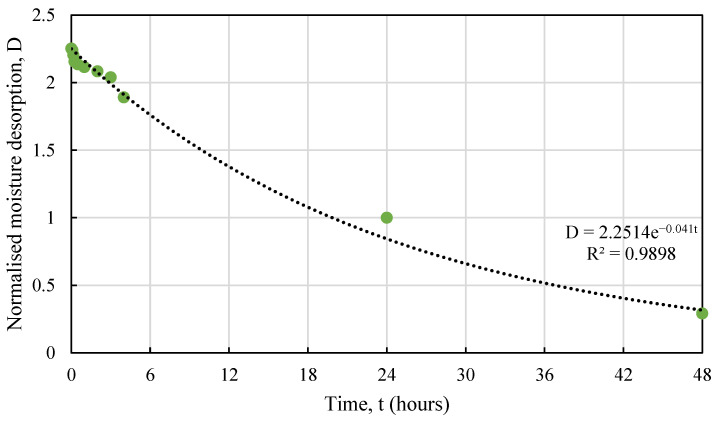
Test vs. proposed relationship for HS3 moisture desorption behaviour.

**Table 2 materials-16-03993-t002:** Proposed parameters for water absorption of HS1, HS2, and HS3.

Shive Size	a	b	R^2^
HS1	0.11	0.75	0.86
HS2	0.12	0.70	0.94
HS3	0.13	0.61	0.97
Average	0.12	0.69	

**Table 3 materials-16-03993-t003:** Desorption of moisture content of HS1, HS2, and HS3 with time from 100% to 93% RH drop.

Time (h)	Moisture Content in Hemp Shives (%)		
	HS1	HS2	HS3
0	375.8	358.8	321.59
1/12	363	353.8	320
1/6	362.6	343.8	314.8
0.25	347.2	336.8	307.8
0.5	340	342.8	305
1	337.4	332	302
2	294.2	305	297.5
3	263.8	242	291.2
24	15.8	23.6	142.8
48	14	14.6	41.5

**Table 4 materials-16-03993-t004:** Proposed constants to predict moisture desorption behaviour of HS1, HS2, and HS3.

Shive Name	*m*	*n*	R^2^
HS1	20.61	−0.078	0.89
HS2	13.75	−0.074	0.93
HS3	2.25	−0.041	0.99

**Table 5 materials-16-03993-t005:** An estimation of hemp shives required for internal curing per unit volume of concrete.

Particle ID	Cement Factor, *C_f_* kg/m^3^ of Concrete	Chemical Shrinkage, *CS* (mL/g)	*α_max_* for W/C = 0.45	*ϕ_HS24h_* (L/kg)	S	D	*M_HS_* (kg)	Hemp as Percentage of Binder mass
HS1	556	0.07	1	2.80	1.09	3	4.25	0.8%
HS2	556	0.07	1	2.70	1.08	2.20	6.08	1.1%
HS3	556	0.07	1	2.52	1.00	0.85	18.14	3.3%

## Data Availability

The data presented in this study are available on request from the corresponding author. The data are not publicly available due to privacy restrictions.

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
