# Peer review of "An Experimental Evaluation of Hemp as an Internal Curing Agent in Concrete Materials"

_materials, 2023, doi:10.3390/ma16113993_

Round 1

Reviewer 1 Report

This manuscript explores the “An Experimental Evaluation of Hemp as an Internal Curing Agent in Concrete Materials”. The manuscript is elaborately described and contextualized with the help of previous and present theoretical background. All the references cited are relevant to this area of research. The methods/analytical study are clearly stated. The result and discussion section are clearly presented. The manuscript needs the following modifications before the acceptance.

1. Arrange the key words in alphabetical order

2. What is the novelty of your research?

3. Combine the introduction section and Literature review section.

4. Materials and Method section is too lengthy.

5.  Compare your results with existing studies.

6. Include more experimental photos.

7. Conclusion: Mention the research recommendations and scope for the future work

 Minor editing of English language required

Author Response

Dear Reviewer, 

We appreciate your time and effort to carefully review the manuscript. Please find our responses to your suggestions in the file attached. 

Reviewer 2 Report

The manuscript, entitled "An Experimental Evaluation of Hemp as an Internal Curing Agent in Concrete Materials," presents an experimental study conducted on the water absorption and desorption of hemp particles. However, the paper needs major revisions before it is processed further. Some comments follow:

Introduction section

"The use of hemp in concrete (hempcrete) is gaining popularity [9–11]." – This is a general statement and doesn't need three citations to support it. Moreover, a clear correlation between the statement and the cited studies cannot be observed. Please provide a quantitative evaluation of each cited study and clearly show its relevance for this manuscript.

 The authors state that the novelty of the study is related to a gap in the literature about using hemp in concrete. However, multiple studies had the same main scope on this topic. Could the authors consider highlighting the novelty of this study? Is it about using Australian-grown hemp (does it have special properties)?

Please improve the introduction section, considering some studies from 2023 and 2022 (currently the literature is old); only two citations to studies from 2022 and one to a study from 2021 are adequate.

Literature review.

What is the difference between an introduction and a literature review? Please combine these two sections.

Through the author's evaluation of or discussion of relevant studies, the introduction or literature review could be improved by introducing a few tables or schematic representations that can summarize the parameters and results of the reviewed literature.

Materials and methods section

How many samples have been used to evaluate the properties of hemp sieve aggregates? Please provide the deviation value for each measurement.

Please write the type of % calculated (mass, weight, or volume); probably these are wt.%, but it must be stated in the article also (Table 1, Figure 8, Figure 14, etc.).

Discussion section. The discussion section is missing. In the discussion section, a clear correspondence and comparison between the results of this study and those in the literature should be provided. Please improve, compare the obtained results with those from the literature, and make qualitative and quantitative appreciations.

Conclusion section: Please improve the conclusions and present them following the main recommendations by the academy of giving the conclusions of the study in points with highlights.

Future directions and limitations: Please provide some future directions and limitations of the study.

English spelling and errors

Please check the entire manuscript for typing errors, for example, kg/m3 in Table 1.

Author Response

(The authors gave the same response as above.)

Reviewer 3 Report

A very good research job. 

Author Response

(The authors gave the same response as above.)

Reviewer 4 Report

The manuscript "An Experimental Evaluation of Hemp as an Internal Curing Agent in Concrete Materials " is a good research. However, if internal curing in concrete is mentioned, mixtures in cementitious matrix materials (mortar or concrete) should be performed. Therefore, it is recommended to perform this experimentation, evaluate internal curing in cementitious matrix mixtures and resubmit this manuscript.

 The following are also recommended:

 Line 44. Use “rice husk ash as supplementary cementitious materials” instead of “rice husk pozzolanic admixture”

 What do you mean when you say that hempcrete is a mixture of lime and hemp chips? is the lime in the mixture or does the hemp have an alkaline treatment?

 Add image of each of the hemp particle size groups

 Add the process used to determine bulk density and moisture content (Described by Amziane).

 Table 1. Use kg/m3 instead of kg/m3

 Why is desorption determined at a relative humidity of 93%?

 Was replacement by volume or weight?

 Why is the density higher with a larger particle size?

 Why do smaller particles replace less concrete?

Author Response

(The authors gave the same response as above.)

Round 2

Reviewer 2 Report

Dear authors,

You have done a great job revising the paper according to my recommendations. I don't have any further suggestions that can improve the quality of the paper.

Best regards,

Author Response

The authors would like to thank the reviewer for the comments and effort to review the paper.

Reviewer 4 Report

I appreciate the comments ans response

-As mentioned in the response, they present the analysis of water absorption and desorption in the hemp shives, specifically focusing on the behavior of the fiber, but not as a curing agent for concrete, as the performance of the fiber within a cementitious mixture and its effect on physical and mechanical properties were not evaluated. If there are no mixture in this manuscript that assess the internal curing effect of hemp, the title should be changed

-In the calculation of hemp shives' density, the moisture condition was the same for all three particle sizes.

- Why do smaller particles replace less concrete? Yes, according to the previous response, the replacement was done by volume. Regardless of the density, the replacement will be the same in terms of occupied space

Author Response

As mentioned in the response, they present the analysis of water absorption and desorption in the hemp shives, specifically focusing on the behavior of the fiber, but not as a curing agent for concrete, as the performance of the fiber within a cementitious mixture and its effect on physical and mechanical properties were not evaluated. If there are no mixture in this manuscript that assess the internal curing effect of hemp, the title should be changed

Response – We understand the reviewer’s stance on the current title of the article, but we would like to make note that this paper deals not only with absorption and desorption of hemp shives, but also presents a careful review about the required characteristics of an internal curing (IC) agent for concrete material. Through this review, the significance of water absorption and desorption behaviour for an IC agent was revealed, hence, such characteristics for hemp shives were investigated as prime focus within the research.

Additionally, based on the absorption-desorption behaviour of hemp shives an analytical model was proposed to predict the mass required when used for internal curing in concrete. The basis for such a model lies in the prior research conducted on IC agents, e.g., lightweight aggregates (LWA). The proposed model is intended to be used in future research while producing concrete or mortars using hemp shives as IC agents.

The current research is not focused on simply characterizing the hemp shives from materialistic perspective, rather emphasises on how it could be used in concrete as IC agent. Thus, we would like to urge the reviewer to reconsider the current title of the manuscript as the best suited one.

-In the calculation of hemp shives' density, the moisture condition was the same for all three particle sizes.

Response –

The process as described by Amziane et.al. was followed while determining the bulk density of the hemp shives. The process initiates with drying the hemp shives samples. Next, the shives were filled to half of a glass cylinder and the height covered by the shives was marked in the cylinder. Then, shive mass was determined. Lastly, water was filled in the cylinder to the previously marked height of the shives in order determine the volume of the shives.

In the method, drying of hemp shives was crucial. Unlike normal aggregates, hemp can absorb moisture from the atmosphere which would alter its density significantly. Thus, it was essential that the moisture condition of all three size ranges in consideration was taken to the same level.

Section 2.2.2 has been rewritten to clarify the process of bulk density determination of hemp shives.

Why do smaller particles replace less concrete? Yes, according to the previous response, the replacement was done by volume. Regardless of the density, the replacement will be the same in terms of occupied space

Response –

Smaller particles do not actually replace less concrete. Rather the required mass for smaller particles that’s HS1 is less compared to larger particles as, HS2 and HS3 to produce same concrete. This was mentioned in Table 5 (page 17) and the mass was determined using equation 7. Underlying cause could be attributed to the varying absorption and desorption characteristics of the hemp shives demonstrated by the respective size ranges. The smallest particles in HS1 had higher absorption capacity and faster desorption rate compared to other particle size groups (refer to Figure 8 and 14). Thus, the water required for internal curing was supplied by less mass of HS1 particles compared to HS2 and HS3 size groups.

Confusion regarding mass or volume replacement may have appeared due the last column of Table 5 titled as, “Hemp as percentage of binder content”. Here, the authors’ used “content” to refer as “mass” rather than volume. Now the column title has been changed as, “Hemp as percentage of binder mass.”